# Loc-FACMAC: Locality Based Factorized Multi-Agent Actor-Critic Algorithm for Cooperative Tasks

## Abstract

In this work, we present a novel cooperative multi-agent reinforcement learning method called **Loc**ality based **Fac**torized **M**ulti-Agent **A**ctor-**C**ritic (Loc-FACMAC). Existing state-of-the-art algorithms, such as FACMAC, rely on global reward information, which may not accurately reflect individual agents' actions' influences in decentralized systems. We integrate the concept of locality into critic learning, where strongly related agents form partitions during training. Agents within the same partition have a greater impact on each other, leading to more precise policy evaluation. Additionally, we construct a dependency graph to capture the relationships between agents, facilitating the partitioning process. This approach mitigates the curse of dimensionality and prevents agents from using irrelevant information. Our method improves upon existing algorithms by focusing on local rewards and leveraging partition-based learning to enhance training efficiency and performance. We evaluate the performance of Loc-FACMAC in two environments: Multi-cartpole and Bounded-Cooperative-Navigation. We explore the impact of partition sizes on the performance and compare the result with baseline MARL algorithms such as LOMAQ, FACMAC, and QMIX. The experiments reveal that, if the locality structure is defined properly, Loc-FACMAC outperforms these baseline algorithms up to 45% , indicating that exploiting the locality structure in the actor-critic framework improves the MARL performance.

## 1 Introduction

Multi-Agent Reinforcement Learning (MARL) is a framework (Foerster et al., 2017; Tan, 1993) that enables a group of agents to learn team behaviors by interacting with an environment. Recently, the impact of MARL has become quite evident in a range of areas (Jiang et al., 2022; Chu et al., 2019; Zhang et al., 2022; He et al., 2016). In the field of reinforcement learning, multi-agent coordination plays a crucial role in various applications such as cooperative searching, human-robot interaction, product delivery, and soccer (Ji et al., 2022; Qie et al., 2019; Vorotnikov et al., 2018; Ota, 2006; Jiménez et al., 2018). In these scenarios, agents often rely on local observations to make decisions that benefit the entire team. In many MARL algorithms, access to global rewards is assumed. However, the assumption does not hold in many scenarios as agents often need to learn cooperative behaviors based only on local observations and local or group rewards. In this paper, we propose a new MARL technique that learns using the locality inherent in many multi-agent coordination scenarios.

Similar to single-agent RL, most existing MARL frameworks can be classified into two categories: *value-based* (Watkins & Dayan, 1992; Sunehag et al., 2017; Rashid et al., 2018; 2020; Son et al., 2019; Kortvelesy & Prorok, 2022; Xu et al., 2021) approaches and *actor-critic* approaches (Konda & Tsitsiklis, 1999; Peng et al., 2021; Wang et al., 2020). In value-based approaches, agents learn to estimate an action-value function by exploring the action space and choosing the action with the maximum action value. Value-based approaches are commonly used in MARL, in part, because QMIX (Rashid et al., 2018) has shown the potential of solving complex coordination problems such as the Star-Craft Multi-Agent Challenge (SMAC) (Samvelyan et al., 2019). The core idea of QMIX is to utilize a monotonic mixer to estimate the joint-action value $Q_{tot}$ from the individual state-action values $Q_i$. The joint-action value function evaluates the agents' performance and

modifies the policy using back-propagation. The idea of QMIX has been extended in several ways, including WQMIX (Rashid et al., 2020) and Qtran (Son et al., 2019).

Although value-based approaches have shown the potential for solving complicated tasks, the curse of dimensionality prevents the approach from being applied to large-scale tasks. For instance, in QMIX-type approaches, when the number of agents increases, the joint-action space exponentially increases. Meanwhile, value-based approaches must compare all action values, requiring significant search time. Therefore, value-based approaches are not an efficient method for training many agents at once.

In contrast, the actor-critic approach directly learns a policy and decides actions from the policy rather than maximizing the Q value function. For instance, MADDPG (Lowe et al., 2017) is one of the classical approaches in this category. The actor in MADDPG learns to generate the optimal action and sends the chosen actions to a critic to evaluate the value of the chosen action. Then, the actor

Table 1: Comparison of different approaches.

| Algorithm | Num. of Mixers | Critic | Actor |
|---|---|---|---|
| QMIX (Rashid et al., 2018) | 1 | ✓ | ✗ |
| LOMAQ (Zohar et al., 2022) | $K$ | ✓ | ✗ |
| FACMAC (Peng et al., 2021) | 1 | ✓ | ✓ |
| Loc-FACMAC (This work) | $K$ | ✓ | ✓ |

adjusts the policy according to the score given by the critic. MADDPG can reduce the training time, but agents update their policy via a separate policy gradient while assuming that the actions of all other agents are fixed. Therefore, the policy commonly falls into the sub-optimal solution. A recently proposed algorithm, FACMAC (Peng et al., 2021), overcomes the limitations of solely value-based and only actor-critic approaches by combining them both. In FACMAC, QMIX is used for the critic update, and a centralized policy gradient is used for the actors during training. FACMAC is shown to outperform QMIX and MADDPG on various tasks within the SMAC environment (Peng et al., 2021).

A common feature of most existing MARL approaches is that they aim to maximize a common global reward. However, in various practical applications, it is possible that one agent's actions may not have any effect on another agent. For example, in a task of surveillance by a group of agents (Kolling & Carpin, 2008), if two agents are quite far from each other, it makes sense to assume that their actions will not affect each other. This idea is explored in a recent paper for value function-based MARL approaches, in which the authors proposed the LOMAQ algorithm (Zohar et al., 2022). LOMAQ presents a multi-mixer approach to accelerate the training process by exploiting the locality of the rewards by defining a partition (a subset of agents) across the network of agents. LOMAQ provides theoretical guarantees under fully observable settings that maximizing the global joint-action value is equivalent to maximizing the action value in each partition. The partition's action value reflects the performance of the partition so each agent can learn a local policy that maximizes the local reward of that partition instead of focusing on maximizing the global reward. The feedback in each partition only updates the most correlated agents' actor-critic networks.

Several works have examined the relationship between agents (HAO et al., 2023). Recent approaches utilize attention mechanisms to determine the weights of graph neural networks (GNNs) (Liu et al., 2019; Li et al., 2021), which connect the agents' actions to cooperative behavior. Alternatively, deep coordination graphs (DCG) (Wang et al., 2022; Böhmer et al., 2020) can be used to model the payoffs between pairs of agents. However, existing works dynamically learn the graph during end-to-end learning, resulting in continuous changes to the graph structure.

In this work, we extend the concept of *locality* to actor-critic methods and introduce a novel locality-based actor-critic approach named **Loc**ality-based **Fac**torized **M**ulti-Agent **A**ctor-**C**ritic Algorithm (Loc-FACMAC) for cooperative MARL. The key characteristics of Loc-FACMAC compared to existing methods are summarized in Table 1. Loc-FACMAC separates the process of constructing the dependency graph from policy learning. Once the dependency graph is established, Loc-FACMAC utilizes it across multiple mixers to compute the local joint-action value in each partition. Both critic and actor can leverage locality information to update the network with accurate policy gradient values. Similar to FACMAC, Loc-FACMAC divides the training process of critics and actors, enabling the critic to precisely evaluate action value quality without being influenced by action choices. The actor learns from the high-quality local action-value function and rapidly converges to an optimal policy.

**Contributions.** We summarize our main contributions as follows.

- We introduce a novel two-stage MARL approach named Loc-FACMAC. In the first stage, we construct a dependency graph based on the variance of agents' performance when they utilize state and action information from other agents. In the second stage, agents learn the local policy within the fixed structure of the dependency graph using an actor-critic approach.

- We evaluate the proposed Loc-FACMAC algorithm in two MARL environments: multi-cartpole and bounded-cooperative-navigation. Our framework demonstrates good performance in solving these tasks and is competitive relative to baseline methods. If a proper dependency graph is defined, our framework can achieve the maximum reward with a competitively short amount of training time.

## 2    Problem Formulation

We model the problem as a decentralized, partially observable Markov decision process (Dec-POMDP). We define the process as a tuple $\mathcal{M} = (N, \mathcal{S}, \mathcal{A}, P, r, \Omega, O, \gamma, \mathcal{G})$. Here, $N$ denotes a finite set of agents, $s \in \mathcal{S}$ is the true joint state, and $\mathcal{A}$ is the joint-action space of all agents. At each time step, if the state is $s \in \mathcal{S}$, and each agent $i$ selects an action from a continuous or discrete action space $\mathcal{A}_i$, then we transition to state $s' \sim P(\cdot|s, a)$, where $P(\cdot|s, a)$ represents the transition kernel from state $s$ to $s'$. We define $r$ as the global reward which depends on the global state and the joint action. The discount factor is denoted by $\gamma$. We note that $\Omega$ is the observation space, which implies that at each time step, agent $i$ can observe partial information $o_i$ sampled from $O_i(s, a) \in \Omega$.

Figure 1: This figure considers a five-agent network and shows how different agents are connected. It also shows two ways of partitioning the network to leverage locality in the learning process. Node 4 and node 5 are away from node 1, node 2, and node 3. Node 4 and node 5 can be grouped separately.

$\mathcal{G} = (\mathcal{V}, \mathcal{E})$ is an undirected graph of agents, namely a dependency graph, where $\mathcal{V} = \{1, 2, ..., N\}$ denotes the node and $\mathcal{E} \subseteq \mathcal{V} \times \mathcal{V}$ is the edge between nodes. The dependency graph is an additional feature added to the original Dec-POMDP in this work. The two agents are correlated if an agent is connected with another agent in the dependency graph, as shown in Fig. 1. Hence, the action of an agent affects its neighbor agent's reward. The goal here is to learn a stochastic policy $\pi_i(a_i|\tau_i)$ or a deterministic policy $\mu_i(\tau_i)$ for each agent $i$ where $\tau_i$ is the local action-observation history $\tau_i \in \mathcal{T} = \Omega \times \mathcal{A}$.

We assume that the dependency graph can be decomposed into a collection of partition $\mathcal{P} = \{J_k\}_{k=1}^{K}$, such that $J_k \cap J_l = \emptyset, \forall k \neq l$ and $\bigcup_k J_k = \mathcal{V}$. The global reward $r$ is expressed by $\{r_1, r_2, ..., r_K\}$, such that $r = \sum_{k=1}^{K} r_k$.

We also define the global action-value function as

$$Q_{tot}(\boldsymbol{\tau}(t), \boldsymbol{a}(t)) = \mathbb{E}\left[\sum_{t=0}^{\infty} \gamma^t r(\boldsymbol{\tau}(t), \boldsymbol{a}(t))\right], \tag{1}$$

where we have $\boldsymbol{\tau}(0) = \boldsymbol{\tau}(t), \boldsymbol{a}(0) = \boldsymbol{a}(t)$. Similarly, we define the local Q function $Q_i(\tau_i(t), a_i(t)) = \mathbb{E}[\sum_{l=0}^{\infty} \gamma^t r_i(\tau_i(t), a_i(t))]$, with $\tau_i(0) = \tau_i(t), a_i(0) = a_i(t)$. We assume the condition $Q_{tot}(s(t), a(t)) = \sum_{i=1}^{N} Q_i(s_i(t), a_i(t))$ holds at any time step which is well-defined in value function based approaches (Rashid et al., 2020; Zohar et al., 2022).

## 3    Proposed Algorithm: Loc-FACMAC

In this section, we present the framework of our new algorithm 2, Loc-FACMAC, which is a multi-mixer actor-critic method that allows the agents to synchronize the policy update while the locality information of the rewards is retained.

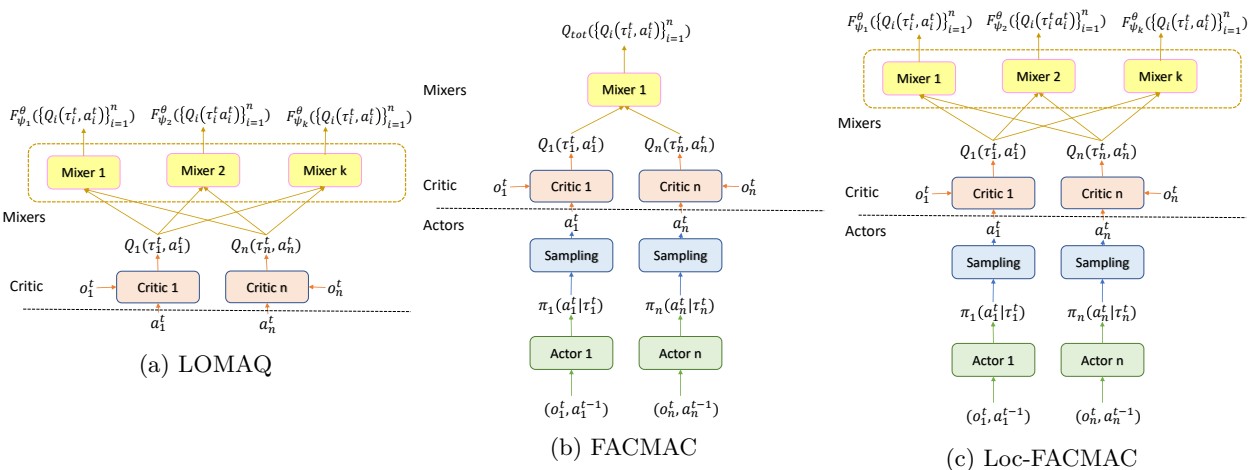

Figure 2: This figure presents the architecture of the proposed LOMAQ, FACMAC, and Loc-FACMAC. Our proposed framework, Loc-FACMAC, consists of Actors, Critics, and Mixers; Actors take local observation and local action history to compute the conditional policy and sample the action from the conditional policy. Critics evaluate the quality of the action. Mixers correlate the local action value and the joint-action value.

---

**Algorithm 1** Proposed Loc-FACMAC Algorithm

---

**Ensure:** $i \in N = 1, 2, ..., n$ are the agents

1: Partition $\mathcal{P}$ of $V$
2: Policy $\pi^i_{\theta_i}(\tau_i)$ with parameter $\theta_i$, where $\tau_i \in \Gamma \equiv (\Omega \times A)$ is local action-observation history
3: Local state $s^i_0 \in \Omega$ drawn from $O(s, i)$, where s is true state of environment
4: Critic Networks $\phi_i \in \mathbb{R}^d$
5: Mixing networks $\psi_k \in \mathbb{R}^e$, where $k = 1, 2, .., K$
6: **for** iter $t = 1, 2, ..., m$ **do**
7:     **for** agent $i = 1, 2, ..., n$ **do**
8:         Sample action $a_i(t)$ from $\pi^i_{\theta_i}(\tau_i)$ and retrieve next observation $s(t + 1)$ and reward $r(t)$
9:     **end for**
10:     We have $Q^\pi_J(\boldsymbol{\tau}^t, \boldsymbol{a}^t, \boldsymbol{s}^t; \phi^t_N, \boldsymbol{\psi}^t_J) = F^t_{\psi_J}(s_t, \{Q^{\pi_i}_i(\tau^t_i, a^t_i; \phi^t_i)\}_{i \in \mathcal{N}(\mathcal{J})})$, $\mathcal{N}(\mathcal{J})$ is the $\kappa$-hop neighbourhood of $J$ in the dependency graph
11:
12:     **Train mixing and Critic network:** update mixing network by minimizing the loss:
13:     $L(\boldsymbol{\phi}, \boldsymbol{\psi}) = E_D[\sum_{J \in \mathcal{P}} (y^{tot}_J - Q^\pi_J(\boldsymbol{\tau}^t, \boldsymbol{a}^t, \boldsymbol{s}^t; \phi^t_N, \boldsymbol{\psi}^t_J))^2]$
14:     $\phi_i \leftarrow \phi_i - \alpha \nabla_{\phi_i} L(\boldsymbol{\phi}, \boldsymbol{\psi})$
15:     $\psi_k \leftarrow \psi_k - \beta \nabla_{\psi_k} L(\boldsymbol{\phi}, \boldsymbol{\psi})$
16:
17:     **Train Actor network:** by update
18:     $\nabla_\theta J(\theta) = E_D[\nabla_\theta \pi \nabla_\pi Q^\pi_{tot}(\boldsymbol{\tau}^t, \pi_1(\tau^t_1), ...\pi_n(\tau^t_n))]$
19:     Update policy parameter $\theta_k \leftarrow \theta_k - \gamma \nabla_\theta J(\theta)$
20: **end for**

---

Loc-FACMAC is built upon FACMAC and overcomes FACMAC's limitation of overgeneralized policy gradients. In FACMAC, the agents' network is updated using the policy gradient computed from the global reward (Peng et al., 2021). Even though the agent does not contribute to the global reward, it still updates its network using the same policy gradient. Using the distorted policy gradient, agents fail to estimate the action value and end up with a sub-optimal policy. Therefore, we adopt the idea of the locality of rewards. The policy gradient is computed from the partitions' reward and used to update the strongly related agents' network. Compared to the policy gradient in FACMAC, the policy gradient computed from partitions' reward is more accurate, so it can give a better update direction to the policy network.

In the Loc-FACMAC, multiple mixers are used to estimate the locality of rewards. Each mixer corresponds to one partition. It takes the local action values and maps them to the partition's joint-action value. By increasing the number of mixers, detailed information on the locality of rewards can be retained. A simple example is estimation four natural numbers $(n1, n2, n3, n4)$ that sum up to a given number. If only the total sum is given, the number of possible combinations could be very large. However, if the subset sums are given, e.g., $n1 + n2$ and $n3 + n4$, the number of possible combinations is reduced. Therefore, the extra information on the locality of rewards can aid agents in finding the globally optimal policy. Then, the loss can be calculated by summing the error in each partition using,

$$L(\phi, \psi) = E_D[\sum_{J \in \mathcal{P}} (y_J^{tot} - Q_J^{\pi}(\tau^t, a^t, s^t; \phi_N^t, \psi_J^t))^2] \tag{2}$$

where $y_J^{tot} = \sum_{j \in J} r_j + \gamma \max_{a^{t+1}} F_{\psi_J}^{t+1}(s_{t+1}, \{Q_i(\tau_i^{t+1}, a_i^{t+1})\}_{i \in \mathcal{N}(\mathcal{J})})$ is the target reward of partition $J$. $\mathcal{N}(\mathcal{J})$ is the $\kappa$-hop neighbourhood of $J$ in the dependency graph.

Loc-FACMAC also inherits the property of FACMAC that it separates the learning process of actor and critic. The framework of Loc-FACMAC consists of two parts: Actor and Mixer. The actor generates an optimal action based on the local observation and historical actions and the mixer criticizes the performance of the agents' action and computes the loss of the network. The advantage of separating the learning process is to prevent the actor from learning from the overestimated/underestimated temporal difference (TD) error. This is critical in MARL because an agent should learn optimal policy by considering the other agents' chosen actions during updating the policy network. However, policy gradient updates the agent's policy network by assuming the other agents' policy is fixed. The assumption fails to be maintained. Especially, when the number of agents increases, the dynamic of agents' behavior becomes hard to predict. Hence, agents frequently get trapped in a sub-optimal policy using MADDPG or QMIX, since no individual agent can modify its optimal action conditioning on fixing all other agents' actions under the sub-optimal policy. This problem is resolved by actor-critic methods in that the mixer first measures the true error of the estimated reward. Then, the actor can be updated by computing the policy gradient from the global joint-action. Since the mixer and actors are updated in two independent steps, agents can synchronize the actor update by considering other agents' actions. The policy gradient is

$$\nabla_\theta J(\theta) = E_D[\nabla_\theta \pi Q_{tot}^{\pi}(\tau^t, \pi_1^t(\tau_1, \theta_1), \pi_2^t(\tau_2, \theta_2), ..., \pi_n^t(\tau_n, \theta_n)))] \tag{3}$$

where $\pi = \pi_1^t(\tau_1, \theta_1), \pi_2^t(\tau_2, \theta_2), ..., \pi_n^t(\tau_n, \theta_n))$ is the collection of all agents' current policy. Using (3), the policy update of an agent does not only rely on its local observation and local action. Instead, the policy gradient is computed using the lasted estimated error from the mixer and the global sampled action $\mu$.

The details of the Loc-FACMAC framework are shown in Algorithm 1. Each agent has its individual actor and critic. The actor takes the inputs of current observation $o_i^t$ and previous action $\mu_i^{t-1}$ to compute conditional policy $\pi_i$. Then, the current action $\mu_i^t$ can be sampled from the policy $\pi_i$ and fed into the critic. The critic produces the utility function $U_i$ which evaluates actions taken by the actor. In the last stage, there are $K$ mixers. Each mixer takes the outputs of the critic and the global state to find a monotonic function $F$ estimating the total rewards of a partition $i$. The number of mixers is equal to the number of partitions $\mathcal{P}$. Each partition contains a subgroup of agents and each agent has to belong to one partition, such that $J_i \cap J_j = \emptyset, \forall i, j \in 1, ..., k$. For example, the set of Partition 2 in Figure 1 is denoted as $J_2 = \{4, 5\}$. If $\kappa = 1$ is selected, the mixer takes the output of $\{2, 3, 4, 5\}$ agents' critic as input to estimate $Q_{J_2}$.

## 3.1 Construct Dependency Graph

The Loc-FACMAC algorithm presented in the previous section assumes that the partitions are already given. In this section, we show how to construct a dependency graph and the partitions.

Before training for the main policy, we first conduct a pretraining process to construct the dependency graph. The dependency graph, which delineates the relationship between agents' actions, is an important component of Loc-FACMAC. The topology of the dependency graph influences both the training efficiency and the performance of the models. An optimal dependency graph manifests as a sparse structure,

connecting only a select few agents of the most significance. Hence, building on the insights of (Wang et al., 2022), we quantify the influence between agents, defined as the discrepancy in agent $i$ with the inclusion of information from agent $j$. This is formalized as:

$$\xi_{ij} = \max_{a_i} \mathrm{Var}_{a_j} \left[ q_i^j(\tau_i, \tau_j, a_i, a_j)) \right]$$

Here, Var indicates the variance function. To obtain the estimation of $q_i^j$, we apply Independent Q-learning to learn the values. An example of a Coupled-Multi-Cart-Pole is shown in Figure 3.

When $\xi_{ij}$ exceeds a threshold, a dependency edge between agent $i$ and agent $j$ is established. This threshold serves as a criterion to determine the significance of the mutual influence between agents. Agents with $\xi_{ij}$ surpassing the threshold are deemed to possess a substantial influence on each other's learning processes, warranting a connection in the dependency graph.

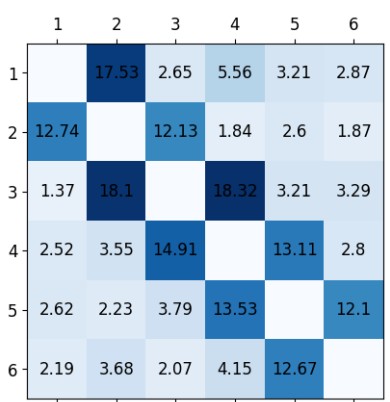

Figure 3: The measurement $\xi_{ij}$ for all pairs of agent in Coupled-Multi-Cart-Pole environment. The higher value indicates a strong relationship between the agents.

## 4 Experiments

In this section, we demonstrate the advantages of our proposed algorithm by comparing it with the other state-of-art MARL algorithms, QMIX (Rashid et al., 2018), FACMAC (Peng et al., 2021), and LOMAQ (Zohar et al., 2022) on two discrete cooperative multi-agent tasks: Coupled-Multi-Cart-Pole and Bounded-Cooperative Navigation. In Coupled-Multi-Cart-Pole and Bounded-Cooperative Navigation, our algorithm outperforms the other tested algorithms. The result also reveals that utilizing multiple mixers can enhance the maximum reward while the actor-critic controls the speed of convergence. Besides that, we study the effect of the dependency graph of agents and the cluster of rewards on the agents' overall performance. The following results will show that by delicately clustering the rewards of strongly related agents, the agents effectively learn a better policy.

To construct the dependency graph, we operate under the assumption that agents share a strong relationship when their geometric distance is relatively small. If a strong relationship exists, it follows that a corresponding edge connecting the agents must be present in the dependency graph.

### 4.1 Coupled-Multi-Cart-Pole

The Coupled-Multi-Cart-Pole problem described in Fig. 4 is developed by Zohar et al. (Zohar et al., 2022) based on the standard Cart-Pole problem. The objective of the Coupled-Multi-Cart-Pole problem is to hold the pole in an upright direction to receive positive rewards. The Coupled-Multi-Cart-Pole problem also restricts the cart movement by linking the cart to its neighboring carts with a spring. Hence, when a cart moves in one direction, its force also impacts the motion of its neighboring carts along the connected spring. Since the cart only impacts its neighboring carts, the relationship can be simply modeled as a linear graph. In the following, we show the simulation result of $n = 6$ carts cooperatively holding the pole for 300 steps, so the ideal maximum reward is 1800.

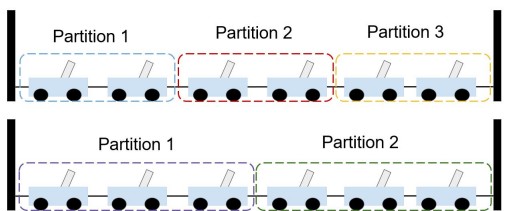

Figure 4: This figure describes the partition for one specific environment of multi-cartpole. We can divide six carts into two possible partitions (2-2-2 and 3-3). There exist other possible partitions, such as 1-2-3.

We apply our approach Loc-FACMAC to compete

with QMIX, FACMAC, and LOMAQ. For Loc-FACMAC and LOMAQ, we set $\kappa = 1$ and partition the rewards into six, $P_i = i, i = 1, 2, ..., 6$ which is the maximum partition.

In this experiment, Loc-FACMAC leverages the locality information and performs the best among all tested frameworks up to 45%. In Fig. 5a, Loc-FACMAC quickly reaches the maximum possible reward at around 140,000 steps. It takes another 150,000 steps to stabilize the rewards. Loc-FACMAC's outstanding performance is due to the integration of the merits of actor-critic and the multiple mixers. The actor-critic method can synchronize the update of agents' policy which guarantees the consistency of the agents' gradient and convergence speed. Adopting this approach, FACMAC converges (at around 250,000) much faster than LOMAQ and Qmix. Loc-FACMAC and FACMAC have a very similar reward pattern in that they both have a high initial momentum boosting the speed of convergence of the two approaches. Meanwhile, the weakness of FACMAC is obvious that FACMAC using a single mixer to coordinate agents overgeneralizes loss between the target global reward and the estimated global reward. The limitation is reflected in the maximum reward of FACMAC which is bounded above by 1250. On the other side, LOMAQ can sustainably improve the policy because of using multiple mixers to capture the detailed locality information. Using multiple mixers can better evaluate the performance of each cluster and hence, a higher reward can be reached. The performance of LOMAQ matches the expectation. LOMAQ has a relatively slower initial value compared to FACMAC and Loc-FACMAC, but it improves the reward sustainably over time and reaches approximately the same reward as FACMAC at 350,000. Loc-FACMAC combines the actor-critic method and multiple mixers so it is the fastest method achieving the highest possible reward among all tested methods and its reward is almost 3 times the fundamental method Qmix at 350,000 steps.

In the aspect of performance, maximizing the number of partitions and the number of mixer inputs can reach the highest performance. However, the trade-off is increasing the computational time. In some machines with limited computational resources, the highest performance may not be their first concern. Therefore, we are also interested in understanding how the size of the partition affects the performance of reinforcement learning agents. In Fig. 5b, we tested LOMAQ and Loc-FACMAC with two different partitions: (1) 6 partitions, {{1},{2},...,{6}} and (2) 3 partitions, {{1,2},{3,4},{5,6}}. For the consistency of the experiment, the inputs of the mixers are always the same in which all agents are taken as the input. The result matches our expectation that if the number of partitions increases, the higher performance can be. The result of Loc-FACMAC using 6 partitions is overwhelming which reaches 1800 at around 140,000. Meanwhile, Loc-FACMAC using 3 partitions performs relatively well compared to LOMAQ using 6 portions and 3 partitions. It obtains roughly 1200 rewards at 350,000 which is slightly better than the number of rewards (1000) of LOMAQ using 6 partitions. Furthermore, in our experimentation, we conducted tests using the LOC-FACMAC algorithm in the environment with 12 agents. The results indicated that increasing the number of partitions in the environment had a positive impact on learning speed.

## 4.2 Bounded-Cooperative-Navigation Environment

The Bounded-Cooperative Navigation task restricts the location of n agents in n bounded regions. The agents can move freely inside their pre-defined region. The bounded regions may have overlapped areas that multiple agents can access the area. The goal of the agents is to cooperatively cover all n landmarks. In this task, the agent's behavior will impact the agents sharing the bounded region. Therefore, if any two agents have a common bounded area, they are connected in the dependency graph. We applied Loc-FACMAC to nine agents' Bounded-Cooperative Navigation task. The number of partitions is 9, $P_i = i, i = 1, 2, ..., 9$ and $\kappa = 1$.

The result in Fig. 5d shows that Loc-FACMAC significantly improves the speed of convergence in the Bounded-Cooperative-Navigation problem. In this problem, the structure of locality dominates the quality of agents' actions. An agent's local reward highly depends on whether its' neighboring agent has already covered a landmark in the common area. Agents can only receive local rewards if its covered landmark is available. Therefore, using only one mixer, the agent is nearly impossible to comprehend its contribution to the local reward because one mixer is not capable of estimating an accurate local reward from computing the global reward. Both Qmix and FACMAC use one mixer so their performance is no different in this task. They both reach approximately 200 rewards at 500,000 steps. In contrast, LOMAQ can reach a higher global reward which obtains 316 rewards at 500,000 steps. It uses multiple mixers to capture the detailed information of the locality so each agent estimates its action value from the perspective of local agents and

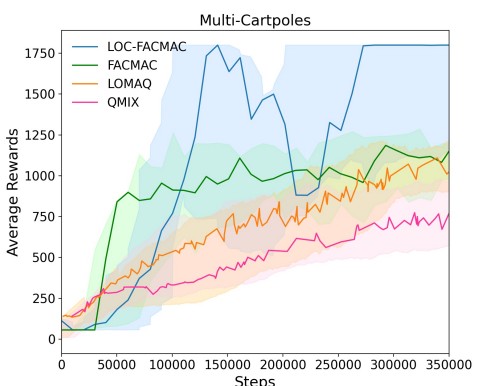
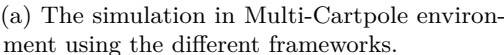

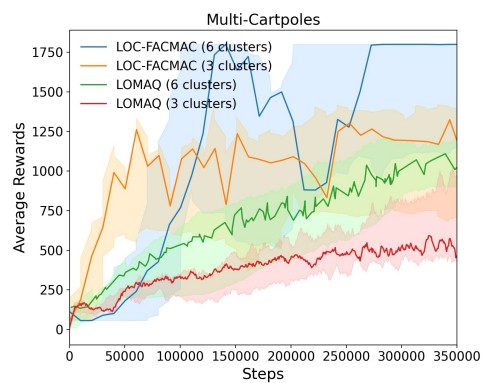

(a) The simulation in Multi-Cartpole environment using the different frameworks.

(b) The simulation in Multi-Cartpole environment comparing the Loc-FACMAC and LOMAQ with different partition sizes.

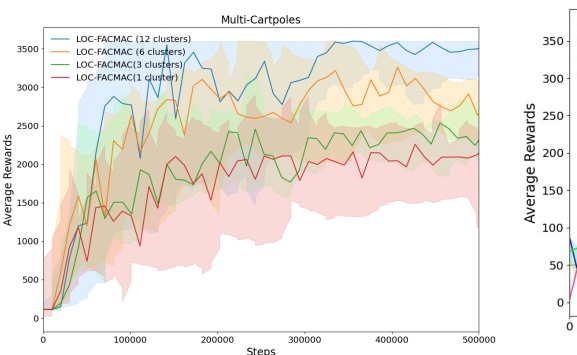

(c) The simulation compares different partitions for one specific environment of the multi-cart pole using Loc-FACMAC.

(d) The simulation in Bounded-Cooperative-Navigation environment using the different frameworks.

Figure 5: This figure compares the performance of the proposed loc-FACMAC algorithm against the baseline algorithms QMIX, LOMAQ, and FACMAC.

global coordination. Similarly, Loc-FACMAC can achieve 300 rewards with a higher speed of convergence. It is due to Loc-FACMAC also utilizing the actor-critic to coordinate the policy update. Hence, it takes much less time to explore the optimal policy for this task.

## 5    Conclusions

In this paper, we introduce Loc-FACMAC, a novel MARL method that combines reward locality with the actor-critic framework to enhance agent performance and learning efficiency. We validate Loc-FACMAC's effectiveness across two tasks, where it outperforms existing methods. Our findings also highlight the correlation between framework structure, maximum reward, and convergence speed. Future research will address challenges such as constructing dependency graphs without prior knowledge and re-evaluating the Individual-Global-Max (IGM) assumption's applicability. Addressing these challenges requires exploring constrained optimization in competitive settings and developing methods to model dynamic agent dependencies without prior assumptions.

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

## Appendix

## A    Background

**QMIX** (Rashid et al., 2018) is a recent framework adopting Centralised Training with Decentralised Execution (CTDE) to solve cooperative MARL tasks. The framework consists of individual agent networks representing the action value functions $Q_i$ and a mixing network that combines the action values of all agents to obtain the global joint-action value $Q_{tot}$. Note that each $Q_i$ is technically a utility function as they do not directly estimate the discounted expected return. In the training stage, the mixing network can access global state information and the global joint-action value. The purpose of the mixing network is to eliminate the difference between $Q_{tot}$ and the function of the action-value set $(Q_1, Q_2, ..., Q_N)$, such that

$$Q_{tot}(\boldsymbol{\tau}, \mathbf{a}) = f_\psi(\boldsymbol{\tau}, Q_1(\tau_1, a_1), Q_2(\tau_2, a_2), Q_N(\tau_N, a_N)),$$

where $\tau_i$ is the local action-observation history of agent $i$ and $a_i$ is the local action specific to agent $i$. Here, $f_\psi$ is required to be a continuous monotonic function to ensure the consistency between the global optimal action and the agents' optimal action. The loss function of QMIX is given by

$$L(\theta, \psi) = \sum_{i=1}^{b} (y_i^{tot} - Q_{tot}(\boldsymbol{\tau}, \mathbf{a}; \theta, \psi))^2, \tag{4}$$

where $b$ is the batch size of transitions sampled from the replay buffer and $y_i^{tot} = r + \gamma \max_{\mathbf{a}'} Q_{tot}(\boldsymbol{\tau}', \mathbf{a}'; \theta^-, \psi^-))$ is the target value of $Q_{tot}$. Here, $\theta$ and $\psi$ are the parameters of the agent's network and the mixing network, respectively. Through back-propagation, $\theta$ and $\psi$ are adjusted to minimize the loss. Accordingly, the learned action-value function $U_i$ can properly evaluate the quality of the action and it only depends on the local observation and the parameter $\theta_i$. In the execution stage, each agent only maximizes its local action value using the local observation.

**LOMAQ** (Zohar et al., 2022) extends the idea of QMIX and improves the algorithm efficiency using the locality of rewards. The core idea is that the agents can be partitioned into different $K$ clusters $\{J_k\}_{k=1}^K$ which defines a partition $\mathcal{P}$, such that $J_k \cap J_l = \emptyset, \forall k \neq l$ and $\bigcup_k J_k = \mathcal{V}$. LOMAQ proved that under the Q-Summation Maximisation (QSM) Condition given by:

$$\max_{\mathbf{a}} \sum_{i=1}^{n} Q_i^\pi(\boldsymbol{\tau}, \mathbf{a}) = \sum_{J \in \mathcal{P}} \max_{\mathbf{a}} \sum_{i \in J} Q_i^\pi(\boldsymbol{\tau}, \mathbf{a}), \tag{5}$$

we can exploit the locality of rewards to reduce the regret of decisions. LOMAQ utilizes $K$ mixers where the value of $K$ can be defined manually to leverage the property of reward locality. Each mixer corresponds to the joint-action value function of one partition $J$ so that the mixer can map the local action values to the joint-action value of the partitions. The loss function of LOMAQ is

$$L_F(\theta, \psi) = \mathbb{E}_{\mathbf{s}, \mathbf{a}, \mathbf{s}'} \left[ \sum_{J \in \mathcal{P}} \left( y_J - F_J^\psi(Q_{i \in J}^\theta(\tau_i, a_i)) \right)^2 \right], \tag{6}$$

where $y_J = \sum_{j \in J} r_j + \gamma \max_{a'} F_J^\psi(Q_{i \in J}^\theta(\tau_i', a_i'))$ is the target reward of partition $J$, and $F_J$ is the joint-action value function of $J$.

**FACMAC** (Peng et al., 2021) is a recent state-of-art framework proposed by Peng et al. (2021). This approach resolves the limitation of biased updates in QMIX by utilizing a factorized critic in an actor-critic framework. Each agent has a critic and an actor. The actor takes the local observation $o_i(t)$ and the past action $a_i(t-1)$ as the input to compute the local policy $\pi_i$. Then, the action $a_i(t)$ is sampled from $\pi_i$. The function of the critic is to evaluate the local action value $Q_i$ from $a_i(t)$ and $\tau_i(t)$. Similar to the structure of QMIX, FACMAC contains a mixer responding to coordinate all agents' actions by mapping $Q_i$ to the joint-action value $Q_{tot}$. Hence, the loss function of the joint-action value function is the same as the loss function in QMIX. Agents' action value functions are updated using the policy gradient given by:

$$\nabla_\theta = \mathbb{E}_{\mathcal{D}}[\nabla_\theta \boldsymbol{\mu} Q_{tot}^\mu(\boldsymbol{\tau}, \mu_1(\tau_1, a_1), \mu_2(\tau_2, a_2), ..., \mu_n(\tau_n, a_n))], \tag{7}$$

where $\mathcal{D}$ is the data buffer and $\theta$ is the parameter of actor network.

The back-propagation of FACMAC relies on a single joint-action value $Q_{tot}$. This means that even if an agent does not contribute to the reward in a particular step, the policy gradient $\nabla_\theta$ still updates its parameters. On the other hand, LOMAQ does not take into account the impact of changing other agents' parameters when updating its parameters, resulting in a slower convergence time. To address these limitations, we propose a novel approach called LOC-FACMAC, which combines the strengths of both methods.

## B  Details of the Environments

**Multi-Cartpole environment** is a variant of Single Cartpole environment. Each agent has to uphold its pole to obtain the reward. Each agent can obtain a +1 reward when its pole does not fall. Multi-Cartople adds a restriction on the Single cart pole that each agent is connected to its neighbor by a spring. Each agent has two inputs that can control its agent to move toward left or right. Its motion affects its neighbor by applying force to the spring. The state of an agent is a four-dimension vector that includes the position, velocity, pole angle, and pole angular velocity. An agent can observe two neighborhood agents' states. The global state is a collection of the state of all agents.

**Bounded-Cooperative-Navigation** is a task in which each agent has to cover a landmark and avoid collision with their agent in its own common area. Each agent is bounded in a circle region and has some areas overlapping with near agents. The reward of an agent is computed by three factors: occupant reward, bonus reward, and collision reward. The occupant reward is triggered when an agent covers the landmark and another agent does not cover the landmark. The value of the bonus reward is based on the distance between the agent and the landmark to enough agent to move toward the landmark. The collision reward punishes the agent when it collides with another agent. The agent can observe the nearby agents' relative position, the landmark's relative position, and the number of occupant landmarks. The state is the collection of agents' velocity, agents' position, and landmarks' position.

## C  Dependency graph

The construction of a dependency graph is important in Loc-FACMAC. If the dependency graph can accurately reflect the relationship of all agents, the network update can target the most correlated agents and improve the efficiency of the training. The construction of a dependency graph requires prior knowledge of the problem. In the graph, each agent is considered as a node. If the agent $i$ is affected by the agent $j$, the node $i$ and the node $j$ should be connected.

The partition in each mixer can be determined manually based on the need of the task and the amount of resources. The smaller size of the partition can provide more locality information and better performance but it also increases the computational time. Also, according to the dependency graph, the strongly related agents should be grouped into one partition. By grouping strong relationship agents into the same partition, the locality of the rewards becomes meaningful because it can truly reflect the performance of the related cooperative agents.

## D  Hyper-parameters

Table 2: Experiment 1:Multi-Cartpole

| Hyperparameter | Values |
|---|---|
| $\epsilon_{start}$ | 1.0 |
| $\epsilon_{end}$ | 0.05 |
| $\epsilon$ anneal time | 50000 |
| batch size | 128 |
| $\gamma$ | 0.99 |
| $\kappa$ | 1 |
| grad norm clip | 20 |
| actor learning rate | 0.0025 |
| mixer and critic learning rate | 0.008 |
| td lambda | 0.2 |
| target update mode | soft |
| target update rate | 0.5 |
| target update interval | 50 |
| Mixer | Linear, Relu, Linear, ReLu, Linear (mixing embed dim 32) |
| Parameter Sharing in Mixer | No |
| Critic | Linear, Linear, Linear (embed dim 64) |
| Actor | Linear, Relu, GRU, Linear (hidden dim 64) |

Table 3: Experiment 2: Bounded-Cooperative-Navigation

| Hyperparameter | Values |
|---|---|
| $\epsilon_{start}$ | 1.0 |
| $\epsilon_{end}$ | 0.05 |
| $\epsilon$ anneal time | 100000 |
| batch size | 128 |
| $\gamma$ | 0.99 |
| $\kappa$ | 1 |
| grad norm clip | 20 |
| actor learning rate | 0.0025 |
| mixer and critic learning rate | 0.0005 |
| td lambda | 0.2 |
| target update mode | soft |
| target update rate | 0.5 |
| target update interval | 50 |
| Mixer | Linear, Relu, Linear, ReLu, Linear (mixing embed dim 32) |
| Parameter Sharing in Mixer | No |
| Critic | Linear, Linear, Linear (embed dim 64) |
| Actor | Linear, Relu, GRU, Linear (hidden dim 64) |

