# OpenReview forum: "Loc-FACMAC: Locality Based Factorized Multi-Agent Actor- Critic Algorithm for Cooperative Tasks"
_TMLR — Rejected by TMLR_

### Review · Reviewer_7UVD · 2024-01-20

**Summary Of Contributions:**

In this paper, the authors argue that one centralized value function cannot assign reasonable credit to each agent, and propose to use the locality of rewards to learn a grouped value function for each group. The idea is combined with an existing algorithm FACMAC, and evaluated on three multiagent environments.

**Audience:**

Yes

**Broader Impact Concerns:**

Not applicable.

**Claims And Evidence:**

No

**Requested Changes:**

1. The authors argue the paper resolves the FACMAC limitation of overgeneralized policy gradient, while the results do not support this claim well. The reviewer suggests providing a toy example (task) with the overgeneralized phenomenon and verifying how Loc-FACMAC resolves it empirically.

2. The reviewer is confused with the partition. From the definition, different groups do not contain the same agent. As in Figure 1, there are two partitions {1,2,3} and {4,5}. Since Agent 2 also connects to Agent 5, which means they are influenced by each other, how does the proposed method consider the connections among different groups?

3. The notations are confusing, for example, in Section 2, the value functions are denoted as $Q_i$, $Q_{tot}$, while in Section 3, $U_i$ is used to represent the utility function, are they the same thing? What's $F$? Is this a mixing network?

4. The Figure 2 is never referenced in the paper.

5. The partition method is not discussed in Section 3. How is it difficult to define the partition for each task is not discussed either. Some papers have already investigated how to measure the relationships among agents automatically, for example [1], the authors should conduct a comprehensive literature review from this direction.

[1] Multi-Agent Game Abstraction via Graph Attention Neural Network. 2020.

**Strengths And Weaknesses:**

Strengths,

1) The idea of learning several grouped value functions based on the locality structure reflects the sparse interaction property in MASs, which makes sense.

2) Experimental results show promising results given pre-defined partitions of agents compared with non-partitioned algorithms.

Weaknesses,

1) The novelty of this paper is limited, as the paper seems like a combination of two existing methods, LOMAQ and FACMAC. The paper simply extends LOMAQ to the actor-critic version.

2) Some technical details are unclear and need to be clarified.

3) Although the limitations of this paper have been discussed, some key aspects are unlisted.

4) The literature review is not extensive and lacks related work to this paper.

---

> ### Author Response · Authors · 2024-03-01
>
> We are thankful to the reviewer for dedicating their valuable time and effort towards evaluating our manuscript, which has allowed us to strengthen the manuscript. We provide thorough response to reviewer’s inquiries in the responses provided below.
>
>
>
> **Question 1:** The reviewer is confused with the partition. From the definition, different groups do not contain the same agent. As in Figure 1, there are two partitions {1,2,3} and {4,5}. Since Agent 2 also connects to Agent 5, which means they are influenced by each other, how does the proposed method consider the connections among different groups?
>
>
> **Response to Question 1:** The construction of partitions directly impacts the mixer. To ensure the sum of the target reward for each partition equals the total target reward $(y_{\text{tot}} = \sum_J y_J)$, the intersection of any two partitions has to be empty.
>
> To address the influence among different partitions, each mixer for a given partition will incorporate additional inputs $(Q_i)$. Let use the example mentioned by the reviewer. Taking the example of Agent 2's connection to partition $\{5,4\}$, the mixer for partition $\{5,4\}$ should take additional input $Q_2$. Moreover, if there is a need to consider indirect influence, additional agents can be included in the input.
>
> we've added more explanations in section 3 to make this clear.
>
> **Question 2:** The notations are confusing, for example, in Section 2, the value functions are denoted as $Q_i$, $Q_{tot}$, while in Section 3, $U$  is used to represent the utility function, are they the same thing? What's $F$? Is this a mixing network?
>
>
>
> **Response to Question 2:** Thank you for pointing out the confusion. We will fix the typo. $Q_i$ and $U$ are equalvant in this paper. And, indeed, $F$ is a mixing network.
>
> **Question 3:** The partition method is not discussed in Section 3. How is it difficult to define the partition for each task is not discussed either. Some papers have already investigated how to measure the relationships among agents automatically, for example [1], the authors should conduct a comprehensive literature review from this direction.
>
>
> **Response to Question 3:**
> In the revised version, we added a new section explaining how we constructed the locality graph. We evaluate the influence between agents using the formula: $\xi_{ij} = \max_{a_i} \text{Var}_{a_j} \left[ q_i^j(\tau_i,\tau_j, a_i,a_j) \right]$. This equation considers the Q-value of agent i and computes the variance based on the state and action information of agent j. A higher value indicates that agent i's action is influenced by agent j. Therefore, we establish a connection between agent i and agent j. Afterward, we utilize Q-learning to estimate this value.

---

> > ### Comment · Reviewer_7UVD · 2024-03-14
> > **Thanks for the response**
> >
> > The reviewer has a follow-up question about the partition. Let's still use the example of two partitions {1,2,3} and {4,5}. In this case, if you input $Q_2$ to the second group mixer, you treat {2,4,5} equally. However, agent 2 connects to agent 5 only. Is there any evidence to show Agent 2 also directly influences Agent 4?  In your explanation, 'if there is a need to consider indirect influence, additional agents can be included in the input.' Seems like you treat the direct influence and indirect influence the same?

---

> > > ### Author Response · Authors · 2024-03-16
> > >
> > > If it is necessary to define direct and indirect influences, consider the example where we set up two partitions: {1,2,3} and {4,5}. In this scenario:
> > >
> > > - The agents in {4,5} are direct influence on the partition.
> > > - The agents in {1,2,3} are indirect influence on the partition.
> > >
> > > The (c, ρ) exponential decay property [1] ensures that the dependence of $Q_{\theta_i}$on other agents diminishes rapidly as the distance between them increases.
> > >
> > > Therefore, according to this property, all agents in the network influence each other, but we can truncate this influence based on the network distance.
> > >
> > > For instance, if we select a 1-hop neighborhood, the inputs for the mixer would include {4,5,2,3}. However, if we extend the neighborhood to a 2-hop distance, the inputs for the mixer would encompass {4,5,1,2,3}.
> > >
> > > [1] Qu, Guannan et al. “Scalable Multi-Agent Reinforcement Learning for Networked Systems with Average Reward.” ArXiv abs/2006.06626 (2020): n. pag.

---

### Review · Reviewer_K9KJ · 2024-01-26

**Summary Of Contributions:**

This work focuses on improving the performance of cooperative multi-agent reinforcement learning. The authors propose adding agent partitioning to the FACMAC algorithm. Essentially combining FACMAC and LOMAQ to support training teams with partition.

**Audience:**

Yes

**Claims And Evidence:**

Yes

**Requested Changes:**

- clarify the contribution and novelty of this work. Try to provide the challenges of proposed solution and how you solve this challenges.

- describe in more detail how the proposed solution work for more complicated environments such as SMAC and demonstration its performance improvement over previous work.

- fix the minor errors mentioned above

**Strengths And Weaknesses:**

Strength:
First of all, the paper is overall well-written and easy to follow. The related works are referenced with good descriptions.

Weakness:
I have a few concerns with this paper:
- The novelty seems limited in this work. The challenge of the paper is not too clear, it is simply combining FACMAC and LOMAQ. What are the challenges for doing agent partitioning on FACMAC?
- In more complicated environments such as SMAC, how do you partition the global rewards into rewards of each partition? Does it require environment support?
- The evaluation results seem to show that the performance improvement is marginal for more complicated environments: For the SMAC benchmark, the proposed algorithm is similar to the baselines.
- Minor error:
1) Figure 4 "(c) he simulation ..." -> "(c) The simulation ..."
2) Figure 4 subfigure (6) caption is about " Bounded-CooperativeNavigation ", but the title is "Multi-Cartpole"

---

> ### Author Response · Authors · 2024-03-01
>
> We extend our sincere appreciation to the reviewer for investing their invaluable time and expertise in evaluating our manuscript. Their constructive critique has significantly contributed to refining our work.Below, we carefully answer the reviewer's questions to make our manuscript stronger and clearer.
>
>
> **Question 1:**
> The novelty seems limited in this work. The challenge of the paper is not too clear, it is simply combining FACMAC and LOMAQ. What are the challenges for doing agent partitioning on FACMAC?
>
> **Response to Question 1:** In the revised version, we added a new section explaining how we constructed the locality graph. We evaluate the influence between agents using the formula: $\xi_{ij} = \max_{a_i} \text{Var}_{a_j} \left[ q_i^j(\tau_i,\tau_j, a_i,a_j) \right]$. This equation considers the Q-value of agent i and computes the variance based on the state and action information of agent j. A higher value indicates that agent i's action is influenced by agent j. Therefore, we establish a connection between agent i and agent j. Afterward, we utilize Q-learning to estimate this value. .
>
> **Question 2:**
> In more complicated environments such as SMAC, how do you partition the global rewards into rewards of each partition? Does it require environment support?
>
>
> **Response to Question 2:**
> SMAC only provides global reward. To apply Loc-FACMAC, we equally distribute the reward to the survived agents.

---

### Review · Reviewer_QLAq · 2024-02-16

**Summary Of Contributions:**

This paper extends the idea of LOMAQ to FACMAC and presents a Locality-based Factorized Multi-Agent Actor-Critic (Loc-FACMAC) framework. The core idea is to manually break the global multiagent learning problem into a set of local multiagent learning problems based on the predefined locality structure. By connecting strongly related agents into partitions and directly utilizing the local summed reward to guide the local critic learning, Loc-FACMAC adopts additional human priori and locality knowledge to help the agents achieve better credit assignments (instead of simply telling the agent a global shared reward).

Experiments on three environments (Multi-cartpole, the StarCraft Multi-Agent Challenge, and Bounded-Cooperative-Navigation) demonstrate that when the locality structure is appropriately defined, Loc-FACMAC can achieve better performance.

**Audience:**

Yes

**Broader Impact Concerns:**

The main concern is that the proposed method is a simple combination of LOMAQ and FACMAC, i.e., replacing the utility function and mixing function of  FACMAC with LOMAQ. The contribution may be very limited.

**Claims And Evidence:**

No

**Requested Changes:**

* Since adopting the manually defined locality structure to boost the learning procedure is the main contribution of LOMAQ, developing methods to model dependency graphs without prior knowledge or assumptions would be a great contribution to this paper.
* Some important related works, which also adopt the locality structure to help do credit assignments or achieve SOTA performance on MARL benchmarks, are missing.
  * Deep coordination graphs, ICML 2020.
  * Context-aware sparse deep coordination graphs, ICLR 2021.
  * Deep Implicit Coordination Graphs for Multi-agent Reinforcement Learning, AAMAS 2021.
  * Boosting Multiagent Reinforcement Learning via Permutation Invariant and Permutation Equivariant Networks, ICLR 2022.
  * Multi-agent game abstraction via graph attention neural network, AAAI 2020.
* The SMAC benchmark is outdated, as many tasks have been shown to be trivially solvable due to the lack of stochasticity in the SMAC benchmark. The multi-agent coordination benchmark proposed in "Context-aware sparse deep coordination graphs" can be used to verify the effectiveness of the locality structure.
* Some experimental settings are not clear.
  * For the Multi-Cartpole environment, since there are 6 carts in total, what's the difference between independent learning (e.g., independent actor-critic or independent Q-learning) and Loc-FACMAC if setting the cluster number also to 6?
  * Since Loc-FACMAC requires the local reward when updating the critic networks, do the authors modify the SMAC environment as it originally only provides a shared global reward?
  * $\mathcal{N}(\mathcal{J})$ is the $\kappa$-hop neighbourhood of $J$. In the experiments, what is the value set for $\kappa$?
  * It’s not clear how many seeds the work used for the main experimental results. The results should be reported across at least 10 seeds.


Minor:
* In the second paragraph, "value-based approaches are commonly used ...". The first letter should be capitalized.

**Strengths And Weaknesses:**

**Strengths:**
* The paper studies an important problem, namely introducing locality to help the MARL algorithms do better credit assignments.
* Most of the paper is well-written and easy to follow.
* Extensive experiments are conducted to show the effectiveness of the proposed method.

**Weaknesses:**
* The proposed method is a simple combination of LOMAQ and FACMAC, i.e., replacing the utility function and mixing function of  FACMAC with LOMAQ. The contribution is very limited.
* Some descriptions are inaccurate. Some words are overclaimed.
  * "So the critic can precisely evaluate the quality of action value `without being influenced by the choice of actions`." Though the critic can use the off-policy learning technique, the values are also influenced by the actions.
  * "The actor learns from the high-quality local action value function and `quickly converges to an optimal policy.`" Can we ensure it's the optimal policy?
  * "Besides that, the actor in the actor-critic approach outputs an action instead of the action value. It is extremely useful when the action space is large since it can `discard the search time` of the highest action value from all actions." If learning a stochastic policy, sampling from the policy costs almost the same time as picking the highest action value.
  * "The actor generates an `optimal action` based on the local observation and historical actions." The 'optimal action' is overclaimed.
* Some symbols are inaccurate and inconsistent, and in some instances, explanations for the symbols are not provided.
  * In Eq.1, the meaning of the bold symbol $\boldsymbol{\tau}$ and $\boldsymbol{a}$ is not given.
  * Below Eq.1, in $\mathbb{E}\left[\sum_{l=0}^{\infty} \gamma^t r_i\left(\tau_i(t), a_i(t)\right)\right]$, $l$ should be $t$.
  * `Eq.3 has some errors.`  Eq.3 should be $\nabla_{\theta_i}=E_D[\nabla_{\theta_i}\pi_i \nabla_{a_i} Q_{tot}(\boldsymbol{\tau}, a_1, \cdots,a_n)|_{a_i=\pi_i(\tau_i, \theta_i)}]$
  * Below Eq.3, $\mu_i^t$ indicates the action of agent $i$ at step $t$. However, in Eq.7, $\nabla_\theta=E_D[\nabla_\theta \mu Q_{\text{tot }}^\mu({\tau}, \mu_1(\tau_1, a_1), \mu_2(\tau_2, a_2), \ldots, \mu_n(\tau_n, a_n))]$, the meaning of $\mu$ is changed and is not given.
* The title of Fig.4d is wrong. 'Multi-Cartpoles' should be 'Bounded-Cooperative-Navigation'
* Some important related works, which also adopt the locality structure to help do credit assignments or achieve SOTA performance on MARL benchmarks, are missing.
  * Deep coordination graphs, ICML 2020.
  * Context-aware sparse deep coordination graphs, ICLR 2021.
  * Deep Implicit Coordination Graphs for Multi-agent Reinforcement Learning, AAMAS 2021.
  * Boosting Multiagent Reinforcement Learning via Permutation Invariant and Permutation Equivariant Networks, ICLR 2022.
  * Multi-agent game abstraction via graph attention neural network, AAAI 2020.

---

> ### Author Response · Authors · 2024-03-01
>
> We express our gratitude to the reviewer for generously dedicating their time and expertise to evaluate our manuscript. Their insightful feedback has been instrumental in enhancing the quality and clarity of our work. In the responses below, we address each of the reviewer's inquiries comprehensively, ensuring that all concerns are adequately addressed and any necessary revisions are implemented.
>
>
> **Question 1:**
>
> For the Multi-Cartpole environment, since there are 6 carts in total, what's the difference between independent learning (e.g., independent actor-critic or independent Q-learning) and Loc-FACMAC if setting the cluster number also to 6?
>
> **Response to Question 1:**
> There are significant differences between independent learning and Loc-FACMAC. In the example you mentioned, within each of the six clusters, a mixer is present to process inputs from the agent and its $\kappa$-hop neighborhood agents, including their actions and states, to compute an output. Consequently, the estimation of local reward by the mixer is influenced by the actions of neighboring agents. In contrast, independent learning does not involve any mixer, and each agent relies solely on its local state for learning.
>
> **Quesiton 2:**
> Since Loc-FACMAC requires the local reward when updating the critic networks, do the authors modify the SMAC environment as it originally only provides a shared global reward?
>
> **Response to Question 2:**
> Indeed, SMAC only provides global reward. To apply Loc-FACMAC, we equally distribute the reward to the survived agents.
>
>
> **Question 3:**
> $N(J)$ is the $\kappa$-hop neighbourhood of $J$. In the experiments, what is the value set for ?
>
>
>
> **Response to Question 3:**
> For simplicity, we choose $\kappa$ as 1.
>
> **Question 4:**
> It’s not clear how many seeds the work used for the main experimental results. The results should be reported across at least 10 seeds.
>
> **Response to Question 4**
> We perform 5 simulations with random seeds for each scenario.

---

### Decision · Action_Editor_12Fp · 2024-04-11

**Recommendation:** Reject

**Comment:**

The reviewers were unanimous in their recommendation of leaning toward reject.

In their official recommendations, one reviewer commented that their main concern is about the contribution. The proposed method is a simple combination of LOMAQ and FACMAC, i.e., replacing the utility function and mixing function of FACMAC with LOMAQ. Also, they stated that the evaluation results seem to show that the performance improvement is very limited. As a result, the claims made in the paper are not well supported by the experimental results.

There was also a specific note about some outstanding problems with the methodology: the authors argue the paper resolves the FACMAC limitation of overgeneralized policy gradient, while the results do not support this claim well. The partition method is not discussed in Section 3. How difficult it is to define the partition for each task is not discussed either. Given the unclarified concerns, the reviewer suggests the authors carefully prepare a new version of the paper and consider re-submitting it again or to other venues.

The last reviewer commented mainly on novelty. As novelty is not a criterion for acceptance at TMLR, in principle this is not a strong determining factor for the decision. However, I would advise the authors to be conservative about claims of novelty and omit any explicit claims of novelty in the list of contributions.

Given the modest gains of the empirical comparison, as a reader I'm left wondering if the added complexity of the Loc-FACMAC is worth the performance. This criticism is reflects on the significance of the paper. I encourage the authors to find a setting where the empirical results more convicingly support the motivation to use Loc-FACMAC.

This was a difficult decision as I believe the paper falls just short of the TMLR acceptance bar. Iencourage the authors to take all the feedback into consideration and resubmit an improved version of the paper with the issues addressed.

**Audience:**

Yes, this topic of this paper is well-suited to TMLR, however it is concerned with a subfield (MARL) of an already subfield (RL) of the main field of machine learning. So the size of the interested audience could be modest.

**Claims And Evidence:**

The claims of contributions listed in the introduction are supported to some degree, but not strongly.

The authors claim that that they introduce a novel approach named Loc-FACMAC. The reviewers agree that this is technically novel, but the novelty is quite limited because it is a straight-forward combination of two existing algorithms.

The second claim is good performance in these tasks and competitive with the baselines, and that with the dependency graph the algorithm can achieve maximum reward in a smaller amount of training time. The performance gain is quite limited and also had some outstanding problems with the methodology. Also the latter part of that claim is not very strongly supported as the results indicate instability in the learning -- and in at least one case dip below the performance of the baseline before surpassing it again (e.g. Fig 5(a)).

More detailed comments are given below.